# Inhibition of Hepatitis E Virus Replication by Novel Inhibitor Targeting Methyltransferase

**DOI:** 10.3390/v14081778

**Published:** 2022-08-15

**Authors:** Preeti Hooda, Meenakshi Chaudhary, Mohammad K. Parvez, Neha Sinha, Deepak Sehgal

**Affiliations:** 1Virology Laboratory, Department of Life Sciences, Shiv Nadar University, Gautam Budh Nagar, Greater Noida 201314, India; 2Department of Pharmacognosy, College of Pharmacy, King Saud University, Riyadh 11451, Saudi Arabia; 3Department of Infectious Diseases and Microbiology, School of Public Health, University of Pittsburgh, Pittsburgh, PA 15261, USA

**Keywords:** Hepatitis E Virus, Methyltransferase, 3-(4-Hydroxyphenyl) propionic acid, antiviral, HEV replication

## Abstract

Hepatitis E Virus (HEV) is a quasi-enveloped virus having a single-stranded, positive-sense RNA genome (~7.2 kb), flanked with a 5′ methylated cap and a 3′ polyadenylated tail. The HEV open reading frame 1 (ORF1) encodes a 186-kDa polyprotein speculated to get processed and produce Methyltransferase (MTase), one of the four essential replication enzymes. In this study, we report the identification of the MTase inhibitor, which may potentially deplete its enzymatic activity, thus causing the cessation of viral replication. Using in silico screening through docking, we identified ten putative compounds, which were tested for their anti-MTase activity. This resulted in the identification of 3-(4-Hydroxyphenyl)propionic acid (HPPA), with an IC_50_ value of 0.932 ± 0.15 μM, which could be perceived as an effective HEV inhibitor. Furthermore, the compound was tested for inhibition of HEV replication in the HEV culture system. The viral RNA copies were markedly decreased from ~3.2 × 10^6^ in untreated cells to ~4.3 × 10^2.8^ copies in 800 μM HPPA treated cells. Therefore, we propose HPPA as a potential drug-like inhibitor against HEV-MTase, which would need further validation through in vivo analysis using animal models and the administration of Pharmacokinetic and Pharmacodynamic (PK/PD) studies.

## 1. Introduction

Hepatitis E Virus (HEV), the causative agent of hepatitis E, is a quasi-enveloped positive-stranded RNA virus with a 7.2 kb genome, and is the only member of the family *Hepeviridae* [1,2]. Hepatitis E generally manifests as self-limited and acute liver disease with higher mortality rates in pregnant women, immune-compromised people, and patients with pre-existing liver diseases [3,4]. Of its four recognized human-infecting genotypes (GT1-GT4), GT1 is mainly transmitted through contaminated drinking water, and GT3 is established as a zoonotic and food-borne infection [2]. The HEV genome encodes for three major ORFs, of which ORF1 encodes the nonstructural proteins of HEV, which are essential for viral replication and invasion of the host immune system. ORF2 encodes the structural protein, i.e., the viral capsid protein of HEV, and ORF3 encodes a small phosphoprotein that helps in the efflux of the virus [5]. In addition, a new coding region, ORF4, has been suggested in GT1 of HEV [6], whose significance in the HEV life cycle remains to be established. ORF1, being the largest ORF of HEV, encodes seven putative/functional domains viz. Methyltransferase (MTase), Y-domain, X-domain, Papain-like cysteine protease (PCP), Hypervariable region (HVR), RNA-Helicase and RNA-dependent RNA polymerase (RdRp) [7]. However, research into whether the ORF1 directly acts as a polyprotein or undergoes proteolytic processing has been inconclusive. Recent studies have reported that the cysteine protease of the virus is responsible for the polyprotein cleavage forming independent functional proteins [8,9].

Eukaryotic Guanylyltransferase (GTase) and MTase activities are responsible for the capping of mRNA and its stabilization by preventing the degradation by endonucleases. The MTase helps transfer the methyl group from host s-adenosyl methionine (SAM) to the mRNA-GMP complex, resulting in m^7^G capped mRNA. HEV-MTase appeared to have some conserved residues like alphaviruses of the MTases family, such as histidine residue, DXXR, and tryptophan residue. Meanwhile, the DXXR motif comprises portions of the SAM binding motif, and the conserved histidine and tryptophan residues are supposed to be responsible for the catalytic activity [10]. The HEV-MTase domain of the virus is responsible for 5′ capping, which is essential for its replication and infection [11]. In addition, HEV-MTase also helps the virus in host immune evasion by antagonizing the effect of type-I IFN response by targeting the RIG-1 activity [12]. Hence, the HEV-MTase could serve as a promising drug target against HEV infection.

In the absence of an approved anti-HEV drug, pegylated-interferon-alpha (PEG-IFN-α) or ribavirin mono/combination therapy is known to be the most effective treatment against HEV infection. However, their severe side effects or drug resistance in a proportion of patients further limit their use. The only vaccine that has been approved in China still awaits global endorsement and access. 

In previous studies, target-based drug screening has been used to treat viral infections such as HCV and dengue virus [13,14]. In this report, we adopted the strategy of the homology modelling of the HEV-MTase to screen and identify some drug-like candidates and their validation using in vitro and in cellulo assays. To confirm the compound’s efficacy, the inhibition of viral replication was validated through a luciferase reporter assay, qPCR, Immunofluorescence Assay (IFA), and Western blot. Using the above parameters, we report a lead molecule, 3-(4-Hydroxyphenyl)propionic acid (HPPA), which interacts and binds with MTase, inhibiting HEV replication. Hence, this study would help to develop new inhibitors or drug-like molecules to target HEV infection.

## 2. Materials and Methods

### 2.1. In Silico Analysis of HEV-MTase

Homology modelling was used to predict the 3D structure of HEV-MTase (aa 34–353) by making use of the structural information from the resolved 3D structure of nonstructural protein1 (nsP1) of the chikungunya virus (CHIKV) (CHIKV-MTase; PDB ID: 6ZOV) [15], and by using I-TASSER (Iterative Threading ASSEmbly Refinement) server [16,17,18]. CHIKV-MTase was selected because this is the closest protein to HEV-MTase, with a known 3D structure. The predicted model was refined by molecular dynamics for 120 ns using the GROMACS (Groningen Machine for Chemical Simulations) software version 5.1 package using the GROMOS96 54a7 force field [19,20,21,22] (MD simulations methodology described in Section 2.5). The Prank Web server (https://prankweb.cz, accessed on 1 March 2022) was used for potential binding site prediction [23,24].

### 2.2. Virtual Screening of the Compounds Library

An in-house database of small molecules of the Natural Product Database (https://zinc12.docking.org/browse/catalogs/natural-products, accessed on 1 March 2022) was constructed from various libraries [25], AfroDb Natural Products [26], Analytic on Discovery NP, Herbal Ingredients In-Vivo Metabolism [27], Herbal Ingredients Targets, IBScreen NP, Indofine Natural Products, NPACT Database [28], Nubbe Natural Products [29], Specs Natural Products, TCM Database @ Taiwan [30] and UEFS Natural Products.

All the ligand 2D structure files (SDF) were first converted to 3D for docking. Ligands were prepared with ligprep and Epik to expand protonation and tautomeric states at 7.0 ± 2.0 pH unit using Epik and Ligprep module of Schrödinger-2019-4 software suite, LLC; New York, NY, 2019 [31]. The low-energy stereoisomers were generated for each ligand, and a few low-energy 3D structures with correct chiralities were retained. The Qikprop module was used to prefilter the ligands to study the significant physiochemically descriptors and pharmacokinetically relevant properties, including absorption, distribution, metabolism, and excretion (ADME) properties. Pre-filtration was done using Lipinski’s rule of five, which helps to ensure the ligand’s drug-like (oral) pharmacokinetic profile [32]. The phase database after ligand preparation consisted of 245,532 compounds (Figure 1). 

The ligprep treated compound library was flexibly docked into the predicted binding site of the HEV-MTase using Grid-based ligand docking with the energetic (Glide) module of Schrödinger-2019-4 software suite, LLC; New York [33,34]. The refined model of HEV-MTase was then docked with substrates SAM and m^7^GTP retrieved from PubChem search to validate the binding site using the SP Glide method. The binding pocket was defined as a set of residues within the radius of 10 Å around the SAM and m^7^GTP bound HEV-MTase model. In the first step, the ligands were rapidly screened by HTVS (High throughput virtual screening); in the second step, the ligands passed out from HTVS were subsequently analyzed in Glide SP (Standard precision) and in the third step, analyzed in Glide XP (Extra precision). A more sophisticated scoring function was applied with each step. HTVS does a crude estimation of ligand-HEV-MTase shape complementarity. The SP method further docked the top 10% scoring ligands from HTVS of the library into the HEV-MTase predicted active site. The top 10% scoring ligands from SP were docked into the predicted HEV-MTase active site by the XP method of Glide. It substantially reduced computational time to screen many compound libraries. Finally, ten compounds targeting specific HEV-MTase protein were selected based on the docking score, interaction of the ligand with predicted active site residues, and commercial availability of compounds for further analysis. Results were analyzed by LigPlot^+^v.2.2 [35] and pymol [36].

To validate the Glide docking results, Blind docking was performed using the AutoDock Vina tool compiled in the PyRx 0.8 virtual screening tool [37,38]. SAM, m^7^GTP, and selected ten compounds were imported into OpenBabel using the PyRx Tool [37]. Using the conjugate gradient algorithm, the energy minimization was performed with the universal force field (UFF). The total number of steps was set to 200, and the number of steps for the update was 1. In addition, the minimization was set to stop at an energy difference of less than 0.1 kcal/mol. The structures were converted to the PDBQT format for docking. The search space encompassed the entire refined 3-D model structure with the following dimensions in Å: center x, y, z = 72.3513, 71.1012, 34.8494 dimensions x, y, z = 58.6406, 52.1740, 65.6434. The docking simulation was run at an exhaustiveness of 8 and set to output only the lowest energy poses. 

### 2.3. Inhibition of HEV-MTase Activity

The MTase coding region (aa 33–353 of pSK-HEV-2 (Accession no. AF444002.1, cDNA clone of Genotype 1) was cloned and expressed in BL21-DE3 cells by following the routine protocol. The recombinant BL21 cells were grown at 37 °C until the optical density (OD_600nm_) reached 0.6 and were induced with 1mM IPTG at 37 °C for 3 h, then harvested and proceeded for solubilization as mentioned earlier [39]. The solubilization of the protein was performed overnight at 4 °C in 10 mM Tris-Cl, pH-8.0, 100 mM NaCl, 1 mM PMSF and 0.5% N-lauryl Sarcosine (NLS, Sigma Aldrich, Burlington, MA, USA). The protein was purified using Immobilized Metal Affinity chromatography (IMAC) and eluted in 10 mM Tris-Cl (pH 8.0), 100 mM NaCl, 200 mM Imidazole and 0.01% N-Lauryl Sarcosine (NLS) [39]. The activity of the purified protein was also studied by varying parameters such as pH, temperature, enzyme concentration, and substrate concentration [39]. The activity was performed in the reaction mixture containing 20 mM Tris buffer, pH 8.0, 50 mM NaCl, 1 mM EDTA, 3 mM MgCl_2_, 0.1 mg/mL BSA and 1 mM dithiothreitol (DTT). Guanosine Triphosphate (GTP) and SAM were used as methyl acceptor and donor, respectively, in the reaction.

The activity for 0.675 μM of purified protein was optimized in the presence of 1 μM SAM (methyl donor) and 0.5 mM GTP (methyl acceptor) at 37 °C for 2 h [39]. For inhibition studies, 1 μM of the ten compounds (synthesized from Molport, USA) were incubated with 0.675 μM of the purified enzyme before adding SAM and GTP, then incubated for 2 h in reaction buffer. The reaction was stopped using 0.5% Trifluoroacetic Acid (TFA) and detected by following the manufacturer’s protocol (MTase-Glo^TM^ Methyltransferase Assay, Promega, Madison, WI, USA). For half-maximal inhibitory concentration (IC_50_) determination, the concentration of HPPA was subsequently increased, and the graph was plotted in terms of percent (%) activity (Y-axis) and log concentration of inhibitor (X-axis), and calculated using GraphPad Prism 9.0.0 (San Diego, CA, USA). The inhibition kinetics were performed in triplicates, and the error bar indicated the standard deviation.

### 2.4. MicroScale Thermophoresis (MST)

The MicroScale Thermophoresis (MST) technique, which was performed to study the binding affinity of HPPA with purified HEV-MTase, was used to study biomolecular interactions. Briefly, the MTase enzyme was fluorescently labelled using a His-Tag Labeling Kit RED-tris-NTA 2nd Generation (NanoTemper Technologies, San Francisco, CA, USA). Sixteen samples of HPPA were prepared in 2-fold serial dilution ranging from 10 μM to 150 pM. The concentration of MTase was kept constant at 50 nM. The ligand (HPPA) was mixed with fluorescently labelled protein at a 1:1 ratio. Subsequently, the samples were loaded in 16 glass capillaries, then loaded into the capillary tray. The data was analyzed using Monolith NT.115 analysis software (NanoTemper Technologies). The experiment was performed as three independent experiments.

### 2.5. Molecular Dynamics Simulations

The binding efficiency of the lead molecule with HEV-MTase was studied using Inhibition assay and MST. The further refined model of apo HEV-MTase and HEV-MTase complex with HPPA was evaluated by molecular dynamics (MD) simulation studies. Initially, the generated model was energy minimized in water using OPLS (optimized potentials for liquid simulations) force field with a convergence threshold of 0.05 by using Protein Preparation Wizard and Macro model module in the Schrodinger-2019-4 software suite to remove steric clashes between atoms and to improve the overall structural quality of predicted models. The refined predicted model of HEV-MTase apo and complex with HPPA were performed using the GROMACS software package using the GROMOS96 54a7 force field [19,20,21,22]. The best pose of each HEV-MTase complex was used for MD simulation studies. The ligand topology was generated by the PRODRG server [40]. Each complex was centered in a dodecahedron box with a minimum edge distance of 10 Å from the edge. The dodecahedron box was solvated with water molecules and 0.15 M NaCl. After system preparation, the system was energy minimized to avoid any steric hindrance by choosing the steepest descent method for a maximum of 5000 steps (Fmax < 100 KJ/mol/nm).

Furthermore, to equilibrate the system at a constant temperature of 300 K in a two-step ensemble process, i.e., NVT and NPT were used for 1000 ps. Finally, all systems were introduced to MD simulation. For analysis, conformations of HEV MTase in each system were studied for the whole 100 ns trajectory. The trajectories were analyzed using g_gyrate, g_hbond, g_rmsf, g_rms, and trajconv tools of GROMACS.

### 2.6. HEV Constructs and In Vitro Transcription 

The plasmid constructs containing full-length HEV genome (pSHEV-3 cDNA clone of HEV genotype 3 accession number-AY575859.1, a kind gift from SU Emerson, NIH, USA) and sub-genomic HEV carrying luciferase gene (GenBank accession number JQ679013) were used to prepare in vitro transcripts. The full-length cDNA and HEV replicon carrying the Renilla luciferase gene was linearized using XbaI and MluI, respectively. The linearized DNA was purified using the column purification method (Qiagen, Germany). Briefly, 5 μg of linearized DNA was used for in vitro transcription [41], and the reaction was performed in 50 μL of reaction mixture containing 1× transcription buffer (Promega, Madison, WI, USA). The reaction mix consisted of 0.5 mM concentration of nucleoside triphosphates (ATP, CTP and UTP and 0.05 mM concentration of GTP (Promega, WI, USA), 5 mM DTT, 2 μL of RNasin (10 U/mL) and T7 RNA polymerase (10U/mL) (Promega, WI, USA). The reaction was performed at 37 °C for 2 h, and 1 μL of T7 RNA polymerase was added after an hour. The RNA was capped using a 0.5 mM Ribom^7^G cap analogue (Promega, Madison, WI, USA), and the transcripts were purified using an RNA cleanup kit (Qiagen, Germany). 

### 2.7. Cell Culture and Transfection 

Huh7 cells (kindly gifted by Shaheed Jameel, ICGEB, India) were cultured in Dulbecco’s modified Eagle’s medium (DMEM) (Gibco™, Life technologies, Grand Island, New York, USA) supplemented with 10% Fetal Bovine Serum (FBS, Qualified, Brazil origin, Gibco™) and 1× Pen-Strep (Gibco™ Penicillin-Streptomycin (10,000 U/mL). The cells were maintained at 37 °C in a humidified incubator supplemented with 5% CO_2_. 

The transfection of full-length in vitro transcribed RNA was performed using electroporation. 5–6 × 10^6^ Huh7 cells were trypsinized and washed with PBS twice. Subsequently, the cells were resuspended in cytomix [42] with 5 mM MgCl_2_ in a 4 mm cuvette and pulsed, as mentioned earlier [42] (Bio-Rad gene pulser, Hercules, CA, USA). After electroporation, the cells were immediately transferred to a 10–12 mL complete medium (DMEM medium with 10% FBS and 1× Pen-Strep) and seeded in six-well plates. After 24 h, the medium was changed, and different concentration of inhibitor, ranging from 0 μM to 800 μM, was added along with the medium. The cells were harvested 7-days post-transfection to check the effect of HPPA on viral RNA replication.

For the HEV replicon luciferase assay, approximately 7 × 10^3^ cells were seeded in each well of the 96-well plate 24 h before transfection. Before transfection, the cells were washed twice with Opti-MEM media (Gibco™, Grand Island, New York, USA) and the transcription mixture was mixed with 400 μL of a lipofectamine-Opti-MEM mixture (Lipofectamine 3000, Invitrogen, Carlsbad, CA, USA) [41]. The mixture was added to the culture monolayer and incubated at RT for 5h. Subsequently, the transfection media was removed and replenished with complete DMEM containing different concentrations of the HPPA ranging from 0 to 800 μM, and further incubated at 37 °C for 72 h. Subsequently, the luciferase assay was performed using the Renilla Luciferase Assay kit (Promega, Madison, WI, USA) as per the manufacturer’s protocol. Before performing the luciferase assay, the media was removed from the cells and washed twice with PBS. The cells were then lysed according to the manufacturer’s protocol, and the luminescence-based assay was performed.

### 2.8. Cell Viability Assay

The cell viability assay was performed using the 3-(4,5-dimethylthiazol-2-yl)-2,5-diphenyl-2H-tetrazolium bromide (MTT) reagent (Himedia Laboratories, India). Briefly, Huh7 cells were seeded in a 96-well plate, and after 24 h, the cells were treated with different concentrations of HPPA, starting from 0 μM to 1 mM. The cells were incubated with HPPA for 72 h, and the MTT assay was performed as per the manufacturer’s protocol (HiMedia Laboratories, India). Cell viability was calculated as a percentage by comparing treated cells with untreated cells. The readings were normalized using blank, containing media and MTT reagent only.

### 2.9. Quantification of Viral RNA

To quantify the viral RNA, the cells (transfected and treated with varying concentrations of HPPA) were harvested 7-days post-transfection and washed twice with PBS to remove any unbound viral RNA. The cell pellet was used to extract the total cellular RNA, isolated from transfected and treated cells using TRIzol (Invitrogen). The RNA was treated with DNase enzyme at 37 °C for 30 min. The DNase-treated RNA was purified using the column purification method (RNeasy MinElute Cleanup Kit, Qiagen, Germany) and proceeded to cDNA synthesis. The cDNA was synthesized using random hexamer primers (Verso cDNA synthesis kit, Invitrogen). The cDNA was taken further for viral copy number quantification using qPCR, which was performed using ORF2-specific primers, and the viral copy number was calculated using a standard equation as described previously [8]. The viral RNA copies were normalized using host GAPDH to equalize the amount of RNA.

### 2.10. Immunofluorescence Assay

An Immunofluorescence assay (IFA) was performed to confirm the effect of the inhibitor on HEV replication. Briefly, Huh7 cells were electroporated with in vitro transcribed HEV RNA, seeded in microchamber slides (Sigma) in complete DMEM Medium (Gibco^TM^) and incubated at 37 °C in a CO_2_ Incubator. After 24 h, the medium was replaced by a DMEM medium containing 0 μM to 800 μM of HPPA. Seven days post-transfection, the cells were washed with PBS and fixed with 4% TFA at RT for 10 min. Next, the cells were washed and permeabilized with PBS containing 0.1% Triton-X, followed by blocking with 3% BSA for 1h at RT. The blocking solution was replaced with an Anti-HEV-ORF2 primary antibody (1:200) and incubated for 1h at 37 °C. The cells were washed thrice with PBS containing 0.05% Tween-20 and set with Alexa flour 488 Anti-Rabbit secondary antibody (1:600). The coverslips carrying the cells were stained with mounting media containing DAPI (Invitrogen). The images were analyzed using Leica Fluorescence Microscope.

### 2.11. Western Blot

The untreated and treated cells were harvested 7-days post-transfection. The protein was extracted from the cells by resuspending them in RIPA buffer (Sigma Aldrich). The protease inhibitor cocktail was added to each sample. The total cell protein was loaded on 10% polyacrylamide gel, and Western blotting was performed [8]. The HEV-ORF-2 epitope-specific antibody was used at a dilution of 1:1000 and incubated for 1 h at 37 °C. The HRP-conjugated anti-rabbit was used as a secondary antibody at dilution of 1:2500. GAPDH (Invitrogen) was used as a loading control at dilution of 1:5000, and the HRP-conjugated anti-mice was used as a secondary antibody at dilution of 1:5000. The blot was developed using Electrochemiluminescence (Bio-Rad, Western ECL substrate). 

### 2.12. Statistical Analysis

All the statistical analysis was performed using GraphPad Prism 9.0.0 (San Diego, CA, USA). The *p*-value < 0.05 was considered statistically significant.

## 3. Results

### 3.1. In Silico Analysis of HEV-MTase

A 3D structure of HEV-MTase was modelled using I-TASSER. Model validation and refinement details are given in Appendix A. To identify potential binding sites, physical descriptors such as pocket rank, score, probability, sas_points, surf_atoms, center_x, y and z and residue numbers were considered (Table 1). An active site (catalytic pocket) rank 1 for substrate SAM and m^7^GTP binding sites in the refined HEV-MTase model is shown in Figure 2. An invariant histidine and tyrosine residues, DXXR Motif residues, are highlighted in Table 1 and Figure 2A,B. The other residues present in the predicted active site cleft are given in Table 1.

### 3.2. Virtual Screening of the Compound Library

SAM and m^7^GTP, the known HEV-MTase substrate, were docked with a refined HEV-MTase structure to validate the docking results. Virtual screening of the compound library under the study is reported in Figure 1, which shows the number of hits that were calculated using different docking protocols of the Glide. The list of the compounds, based on the docking scores (Glide and Vina scores) docked with the refined HEV-MTase model, is shown in Table 2.

The docking study indicated that m^7^GTP and SAM showed hydrogen-bonding interactions with the refined HEV-MTase. In addition, the key interacting residues H32, Asp 81, Arg84 and Tyr196, were conserved within the active site of the refined HEV-MTase (Figure 2C–E).

The m^7^GTP binding site was formed by residues close to the base moiety binding position (Asp81, Try86, Asp135 and Tyr196) and positively charged and polar residues near the triphosphate position (His32, Tyr131, His 134, Tyr188, Asn197, His198, and two non-polar residues (Ala156, Val 186) (Figure 2D). The SAM binding pocket is defined by the residues shared with the m^7^GTP binding site (Asp 81, Tyr131, His134, Asp135, Asn197, His198) and others exclusive for SAM binding (Arg 84, Leu75, Arg76, Pro77, Ser132, Phe74, Asn202) (Figure 2D).

Sequence alignment using BioEdit (RRID: SCR_007361) program showed the conserved residues of CHIKV MTase and SFV-nsP1 or alphavirus-like superfamily NSPs are represented in Figure 2E and Appendix A. Sequence alignment shows potential conservation of residues between the alpha-like MTase domain [15,43,44]. Furthermore, a site-directed mutation study in SFV-nsp1 destroyed or significantly reduced its MTase and GTase activity, confirming that methylation of GTP is an essential prerequisite for synthesizing the covalent guanylate complex [43,44]. 

Moreover, the molecular weights and the log values of these ten compounds were consistent with the guidelines for selecting orally available drug-like molecules [45] and with the acceptable range of Lipinski’s rule of five for drug-like molecules [32], as shown in Appendix A. The 2-D structures of the selected ten potential inhibitors of the HEV-MTase enzyme are shown in Figure 3, and the IUPAC Name of the selected ten compounds are shown in Appendix A.

HEV-MTase complex with HPPA interacted directly with the HEV-MTase residues like m^7^GTP and SAM (Figure 4). Therein, the HPPA was stabilized by conserved His32 and Tyr196 residues. The key interacting residues His32, Asp81, Arg84, and Tyr196, were observed within the binding site (Figure 4).

### 3.3. Cell-Free Inhibition Assay

The HEV-MTase was expressed and purified, and various parameters for MTase activity such as substrate concentration, pH, temperature, and time have been optimized in our recent study [39]. The effect of different inhibitors shortlisted through virtual screening (Appendix A) was studied on the MTase activity of the purified enzyme. The inhibitors were custom synthesized and analyzed for their inhibition of the enzyme activity with proper controls. Compound 1, HPPA, was found to show maximum inhibition among all the inhibitors (Figure 5A). The IC_50_ of HPPA (Compound 1) was determined by varying the concentration of the ligand from 0 to 10 μM in the reaction mixture. It has been seen to inhibit the MTase activity of the enzyme with an IC_50_ Value of 0.932 ± 0.15 μM (Figure 5B).

### 3.4. Binding Affinity of HPPA with HEV-MTase

Further, the binding affinity of HPPA with HEV-MTase was studied using MicroScale Thermophoresis (MST). The enzyme concentration was kept constant at 50 nM, and the ligand or HPPA was varied from 10 μM to 150 pM. The graph was plotted using HPPA concentration on X-axis and normalized fluorescence on Y-axis. The analysis was performed using MO control and MO analysis software. The calculated dissociation constant (_Kd_) of HEV-MTase with HPPA was found to be ~2 μM (Figure 6). 

### 3.5. Molecular Dynamics Simulation

The Molecular Dynamic studies for each system were computed considering 100 ns trajectories. For HEV-MTase RMSD calculation, the Cα backbone was selected where all the systems showed minimal deviations and remained stable during the simulation period. The average RMSD for Apo-MTase and MTase-compound 1 (HPPA) complex was 0.69 nm and 0.55 nm, respectively. Apo-MTase and the inhibitor-MTase system attained equilibrium and got stabilized after 80 ns. The RMSD of the Inhibitor-MTase complex was less than the Apo-MTase, suggesting stable conformation of MTase upon binding to inhibitors (Figure 7). The RMSD of the ligand was also calculated to observe the changes in the binding pattern. The average RMSD of HPPA was 0.40 nm (Figure 7). Since all the systems attain equilibration after 80 ns of simulation, further stability analysis for the complex was carried over for the last 20 ns.

The RMSF values were plotted to represent the Apo-MTase and the HPPA-MTase complex fluctuations. The average RMSF values for Apo-MTase and MTase-HPPA complex were 0.28 nm and 0.32 nm, respectively (Figure 7). Compared to Apo-MTase, the RMSF of MTase-HPPA complex was minimum, suggesting stable binding. Since protein compactness (Rg) is a significant factor in determining the folded state, Rg was calculated to measure the compactness of HEV-MTase. The estimated average Rg for Apo-MTase and MTase-HPPA complex was found to be 2.19 nm and 2.22 nm, respectively (Figure 7). Compactness analysis suggested that identified compounds formed attractive and stable interactions with the modelled MTase.

Moreover, because the intermolecular polar interaction between protein and ligand plays a crucial role in the binding affinity, the hydrogen bond between HEV-MTase residues and identified compounds was calculated for 100 ns of simulation. As a result, the maximum number of hydrogen bonds in the MTase-HPPA complex was 7 (Figure 7). Further, the calculation of protein stability and folding during the simulation demonstrated a lower SASA value for the MTase-Apo than MTase-HPPA.

### 3.6. Effect of HPPA on HEV Replicon Carrying Renilla Luciferase Gene

Out of 10 compounds, Compounds 1, 2, 3, 4 and 6 inhibited the MTase activity. Out of these compounds, only compounds 1 and 4 showed the inhibition of HEV Replicon, but due to the higher cytotoxicity of compound 4 on Huh 7 cells (CC_50_ ~ 10 μM), the compound was discarded. HPPA was selected as a potential antiviral from a cell-free based assay and taken further to study its cytotoxicity and dose-dependent inhibition on p6 HEV-Luc Replicon. To demonstrate that, transfected Huh7 cells were treated with varying concentrations of HPPA ranging from 0 to 800 μM and incubated for 72 h. We have seen that HPPA treatment significantly reduced the viral-replication-dependent luciferase activity (Figure 8). Furthermore, the 50% cytotoxic concentration (CC_50_) and 50% inhibitory concentration (IC_50_) of the compound were found to be more than 1 mM and 77.12 ± 18.7 μM, respectively. At 77.12 ± 18.7 μM, approximately 95% of cells were viable. The IC_50_ of the compound was determined using GraphPad prism 9.0.0. Both the experiments were performed in triplicates as three independent experiments.

### 3.7. Inhibition of HEV Replication Validated Using qPCR, Western Blot, and Immunofluorescence Assay

HPPA inhibits viral replication in luminescence-based assay without any cytotoxic effects. HPPA was also examined further to check the effect on HEV RNA levels in vitro. The impact of HPPA on HEV replication was also validated using western blot and immunofluorescence assay. The HEV RNA quantification was performed using ORF2 primers. The viral RNA copies were found to decrease from ~3.2 × 10^6^ in untreated or mock samples compared to ~4.3 × 10^2.8^ copies in 800 μM of HPPA (Figure 9A). The graph represents the log_10_ RNA copy number per μg of total cellular RNA on the Y-axis.

Similarly, the effect of the compound was seen in mock and treated cells through immunofluorescence and western blot using HEV-ORF2 epitope-specific antibody (raised against ORF2 epitope QQDKGIAIPHDIDLC by GenScript) [9]. The decrease in ORF2 expression was seen when compared to the untreated level using Western blot (Figure 9B). The expression of ORF2 was also seen using HEV-ORF2 specific antibody detected through Alexa fluor 488 secondary antibody (Figure 9C). Again, a decrease in fluorescence was seen compared to the untreated panel. The total fluorescence was quantified using Image J, and the fluorescence level was decreased from 100% to ~25% in the 800 μM treated sample (data not shown).

## 4. Discussion

Viral MTases participate in the capping of the 5′ end of mRNA, leading to an essential structural modification that limits RNA degradation by 5′–3′ exoribonucleases and allows uninhibited viral replication. HEV is genetically close to the alphavirus-like super family plus-strand RNA viruses, including brome mosaic virus (BMV) [46], tobacco mosaic virus (TMV) [47], and bamboo mosaic virus [48], which bear a signature of MTase domain. HEV-MTase, a key enzyme of virus replication, is an understudied enzyme whose presence, characteristics, and functional aspects remain established. 

In a previous study on HEV-RNA capping, Magden et al. [49] expressed a long stretch of HEV Genome (1–979 aa), expressing a protein of 110-kDa, which was reported to have GTase and MTase activity. Since the construct was too long, the enzymes could not be individually mapped on the viral genome, limiting the Biochemical and functional characterization. In the present study, we selected a smaller HEV genome sequence of 33–353 aa depicted as the MTase region by Emerson et al. [11]. In a recent study, this region of approximately 37-kDa covering a putative MTase domain was active and characterized biochemically and biophysically [39]. The same sized fragment giving MTase activity was seen when HEV ORF1 was cleaved using Baculovirus expressed HEV-PCP [9]. Also, HEV polyprotein processing was studied using Huh 7 cells using the BacMam system [8]. Conservation of residues essential for both the reactions in the alphavirus-like superfamily implies that they use a capping mechanism like the alphaviruses. Our amino acid sequence alignment of CHIKV-MTase and SFV-nsp1 and alphavirus-like superfamily HEV-MTase placed these conserved residues at positions corresponding to Ado-MTase binding sites of cellular MTase, suggesting that they all may be structurally related.

Having established the authenticity of the enzyme, we wanted to study its feasibility as a drug target and identify potent inhibitors that will enable the arrest of HEV replication. Earlier, the identification of target-based antiviral agents has successfully treated viral infections, such as Zika virus, HCV, dengue virus, etc. [50,51,52]. In the absence of a crystal structure, we modelled a 3D design of HEV-MTase. Although sequence similarity with the template used for homology modelling was low (34%), conservation of key active site residues were observed. Furthermore, the predicted structural model could be used for in silico screening of compound libraries to identify HEV-MTase inhibitors. The most promising HEV-MTase inhibitor (HPPA) identified through virtual screening is a flavonoid metabolite of *Clostridium orbiscindens,* which is present in some human guts and prevents Influenza virus propagation [53]. The compound inhibited the enzymatic activity of the purified HEV-MTase (Figure 5) and showed a binding affinity with HEV-MTase studied using MST (Figure 6). The in vitro studies were coupled with the HEV replication system in mammalian cells. The compound has been further analyzed to see its effect on viral replication, validated through qPCR, Western blot, and IFA (Figure 9). Since the homology between the human and HEV-MTase is significantly less, it is unlikely that HPPA will affect the host MTase; however, experimental validation to check the effect of HPPA on host mRNA capping will be the future scope of this study. Nonetheless, the pharmacodynamic testing of HPPA in animals and further clinical trials will be required to evaluate its efficacy towards development as a novel anti-HEV drug. 

## Figures and Tables

**Figure 1 viruses-14-01778-f001:**
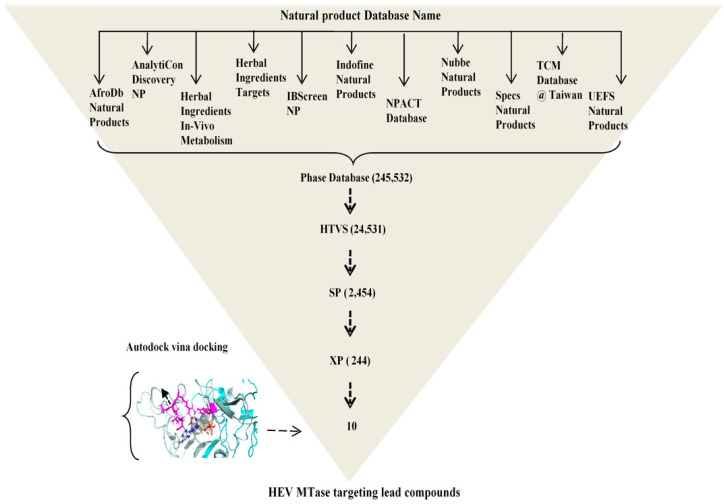
A flow chart depicting protocol for Virtual Screening against HEV-MTase.

**Figure 2 viruses-14-01778-f002:**
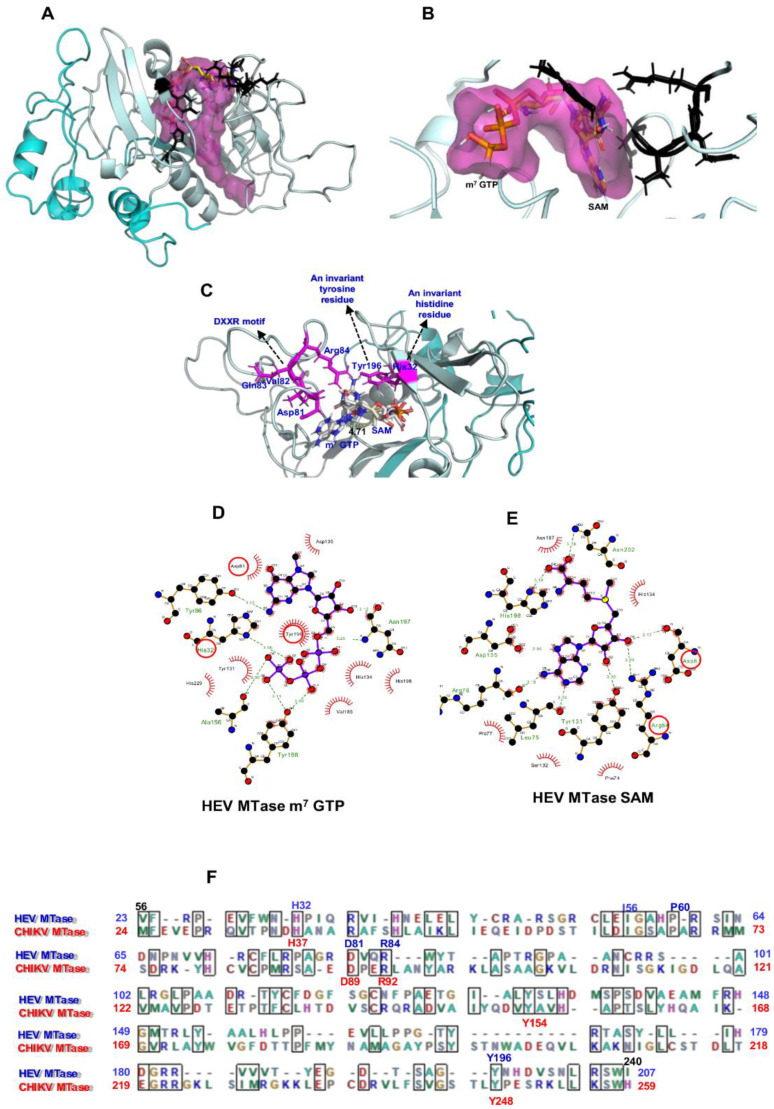
Computational docking of m^7^ GTP and SAM on HEV-MTase model. (**A**) 3D structure model of HEV-MTase protein (region 34–353) after MD simulations; Putative MTase region 60 to 240 amino acid is highlighted in light cyan, and 34–59 to 241–353 amino acid regions are in dark cyan. The Surface view of the ligand-binding site in the HEV-MTase model is highlighted in dark pink. (**B**) For clarity, only the first rank pocket in the structure of HEV-MTase is shown here. Docked poses of m^7^ GTP and SAM are displayed in the binding site of putative HEV-MTase protein. The active site residues are shown as balls and stick with a black colour code. (**C**) Docked poses of m^7^ GTP and SAM (grey carbon, blue nitrogen, red oxygen, yellow sulfur, and phosphate in pink) are displayed in the binding site of the putative HEV-MTase region. The docked poses show distance (4.71Å) as yellow dotted lines between a carbon atom of methyl group donor and N7 methyl group acceptor in the m^7^GTP and SAM binding site of HEV-MTase protein. An invariant histidine 65 residues (H32), a DXXR Motif residues Asp114, Val115, Gln116 and Arg117 (Asp81, Arg84), and the invariant tyrosine 229 residue (Tyr196) in the HEV-MTase complex with m^7^GTP and SAM are represented in red colour circles. (**D**,**E**) The 2D interaction diagrams. The docked poses display interactions in the m^7^GTP and SAM binding site of the HEV-MTase protein. In the charts, residues highlighted with a red circle are common in the HEV, CHIKV and SFV MTase proteins. Conserved residues His32, Asp81, Arg84, and Tyr196 were previously mutated in the SFV MTase site-directed mutagenesis study. Sidechains of interacting residues of the HEV-MTase protein are shown as sticks in pink colour. (**F**) Amino acid Sequence alignment of HEV-MTase (23 to 207 correspond to (Putative HEV-MTase 56 to240 region) and CHIKV-MTase domain 24 to 259 region. Conserved residues of HEV-MTase and CHIKV-MTase are highlighted in blue and red colours, respectively.

**Figure 3 viruses-14-01778-f003:**
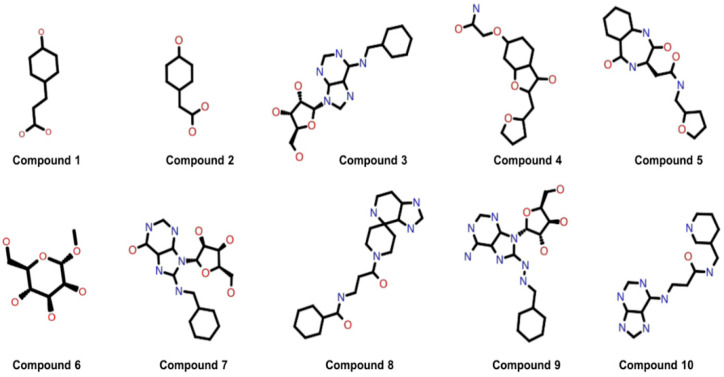
2D structure of the best ten compounds identified through the refined model structure of the HEV-MTase docking study.

**Figure 4 viruses-14-01778-f004:**
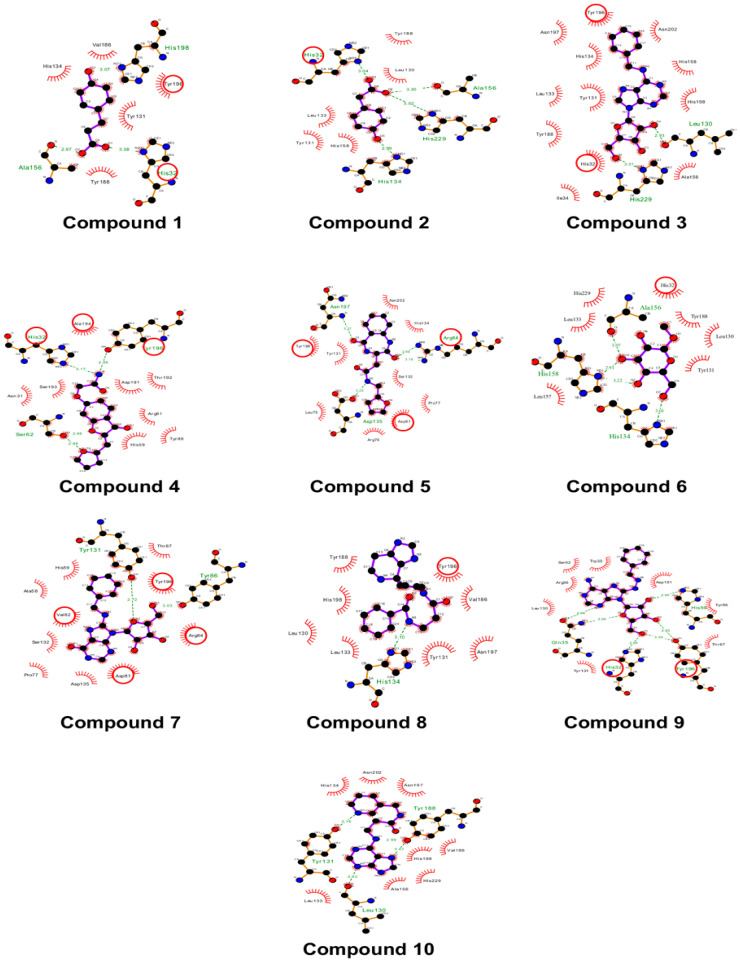
The 2D interaction diagram. The docked poses display interactions in the SAM and m^7^GTP binding site of the predicted model of HEV-MTase protein. In the diagrams, residues highlighted in the red circle are common SAM and m^7^GTP binding residues in the HEV-MTase complexes. Sidechains of interacting residues of the HEV-MTase protein are shown as sticks in pink colour.

**Figure 5 viruses-14-01778-f005:**
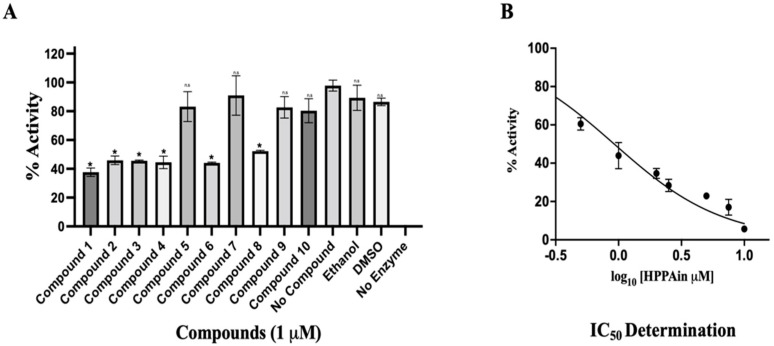
Cell-free inhibition assay and IC_50_ determination of HPPA: (**A**) The graph represents the % of HEV-MTase activity in different compounds concerning compound-free control. The no compound control was taken as 100% MTase activity, and no enzyme control was taken as 0% activity of HEV-MTase. The inhibition assay was performed in triplicates, and the error bar indicates the standard deviation. The statistical analysis was performed using a student *t*-test, and a *p*-value < 0.005 was considered statistically significant. * *p* < 0.0001; n.s *p* > 0.005. (**B**) The graph represents the mean value of triplicate measurements for IC_50_ determination. GraphPad Prism was determined by fitting the curve using log(inhibitor) vs. normalized response. The error bar indicates the standard deviation. The determined IC_50_ for HPPA was 0.932 ± 0.15 μM.

**Figure 6 viruses-14-01778-f006:**
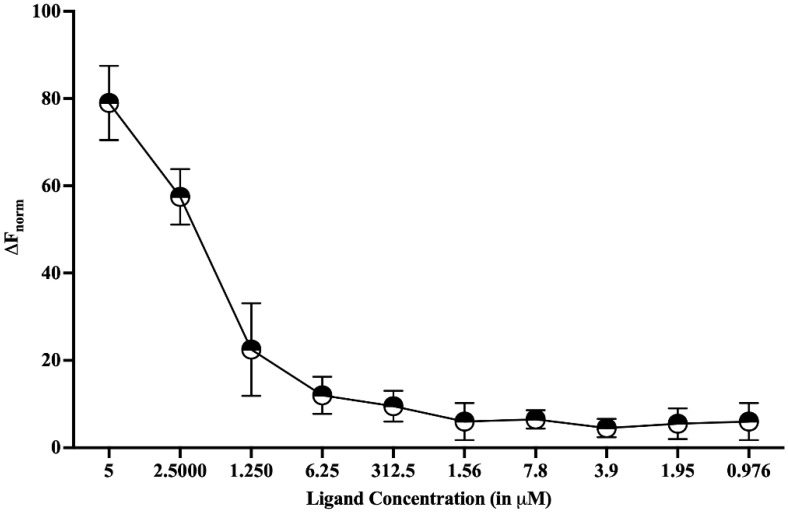
MicroScale Thermophoresis (MST). Dose-response curve to study the binding affinity between HEV-MTase and HPPA was plotted. The graph’s X-axis represents the concentration of ligand (HPPA), and the Y-axis represents the % fluorescence change. The calculated K_d_ for HEV-MTase and HPPA was approximately 2 μM. The graph describes the mean value of three independent experiments, and the error bar indicates the standard deviation.

**Figure 7 viruses-14-01778-f007:**
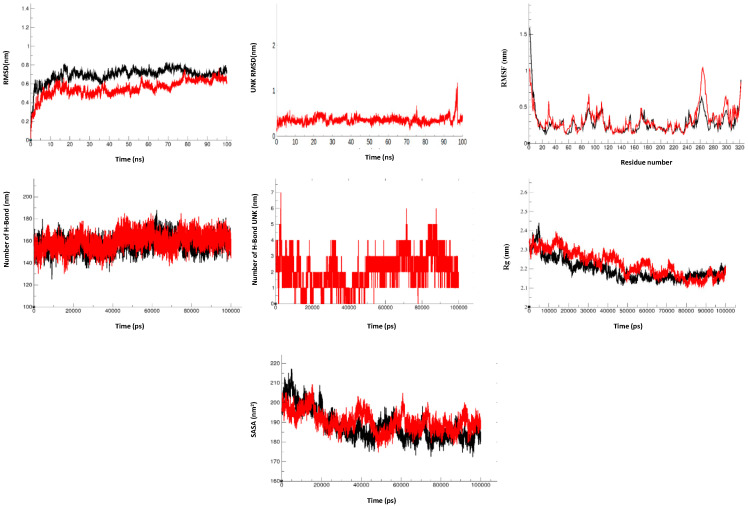
Molecular dynamics simulations of HPPA with HEV-MTase: RMSD of the Cα backbone of apo-MTase and its complexes with respect to time (100 ns), RMSD of compounds over 100 ns, RMSF of Ca atoms of MTase complexes over the 100 ns. Stability of MTase from HEV-MTase: The number of hydrogen bond interactions between MTase and compounds during simulation are shown. The radius of gyration (Rg) of MTase and in complex with compounds over the simulation. Total solvent accessible area (SASA) with respect to time is shown. The apo-MTase (black) and its complex with inhibitor HPPA (red) are represented.

**Figure 8 viruses-14-01778-f008:**
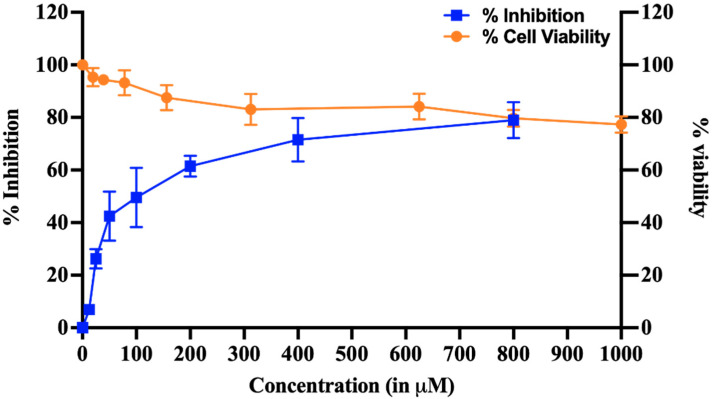
Cell Viability and inhibition of HEV Replicon. The dose-dependent curve represents % inhibition of viral replication in HEV replicon-based assay and cytotoxicity profile of HPPA. The dose-dependent graph was generated by quantifying luminescence, and cytotoxicity was examined using an MTT assay. The blue line in the graph represents the % inhibition, and the orange line represents the cell viability in the presence of HPPA. The calculated CC_50_ and IC_50_ of the compound were found to be more than 1 mM and 77.12 ± 18.7 μM, respectively.

**Figure 9 viruses-14-01778-f009:**
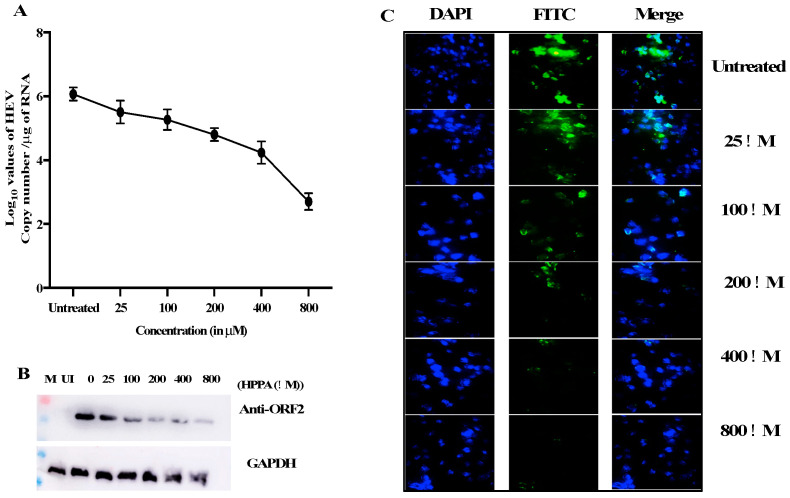
Inhibition of HEV Replication. (**A**) The RNA copy number of HEV was quantified using qPCR. HPPA reduced RNA levels from 3.2 × 10^6^ copies to 4.3 × 10^2.8^ copies per μg of total cellular RNA. The graph represents the log_10_ RNA copy number per μg of total cellular RNA. (**B**) The ORF2 level was checked using western blot in mock and treated cells, and a decrease in ORF2 expression was seen compared to the untreated group. (**C**) The expression of ORF2 was also seen using HEV-ORF2 specific antibody detected through Alexa fluor 488 secondary antibody. The decrease in fluorescence was seen when compared to the untreated panel. The fluorescence was quantified using Image J, and the fluorescence level was decreased from 100% in untreated cells to ~25% in an 800 μM treated sample.

**Table 1 viruses-14-01778-t001:** Prediction of ligand binding site in HEV-MTase refined model structure using Prank Web server. The invariant tyrosine, histidine, and DXXR Motif residues are represented in bold residues numbers in the HEV-MTase model.

Pocket Rank	Score	Probability	Sas_points	Surf_atoms	Center_x	Center_y	Center_z	Residue Number
1	78.45	0.993	364	156	77.6652	73.8546	28.7468	Thr126, Ile128, Leu130, Tyr131, Ser132, Leu133, His134, Asp135, Ala156, His158, Val186, Tyr188, Asp191, Thr192, Ser193, Ala194, Gly195, **Tyr196**, Asn197, His198, Asp199, Asn202, Leu203, His229, Trp30, Asn31, **His32**, Gln35, Arg36, His39, Phe4, Leu42, Glu43, Cys46, Arg47, Ala48, Cys53, Glu55, Gly57, Ala58, His59, Pro60, Ser62, Ile63, Asn64, Asp65, Asn66, Asn68, Val70, Phe74, Leu75, Arg76, Pro77, **Asp81 Val82**, **Arg84**, Tyr86, and Thr87.
2	2.5	0.07	12	14	76.3124	68.2785	45.3086	Leu165, Tyr170, Tyr175, Gly227, Cys228, Val256, Ser258, and Phe260.
3	0.82	0.004	9	9	79.5281	80.4112	19.0718	Phe114, Phe74, Pro77, Ala78, Val82, and Ala94.
4	0.73	0.002	3	3	66.31	77.96	32.2667	Val141, Ala155, and Leu157.

**Table 2 viruses-14-01778-t002:** HEV-MTase and compounds docking affinity (kcal/mol).

Compound NAME	Glide Docking Affinity (kcal/mol)	Autodock Vina Blind Docking Affinity (kcal/mol)
M^7^GTP	−6	−8
SAM	−5	−7
Compound 1	−6	−7
Compound 2	−5	−6
Compound 3	−7	−10
Compound 4	−6	−7
Compound 5	−6	−6
Compound 6	−5	−6
Compound 7	−7	−8
Compound 8	−6	−8
Compound 9	−5	−8
Compound 10	−6	−8

## Data Availability

Not applicable.

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
