# Peer review of "Inhibition of Hepatitis E Virus Replication by Novel Inhibitor Targeting Methyltransferase"

_viruses, 2022, doi:10.3390/v14081778_

Round 1

Reviewer 1 Report

Hepatitis E virus poses a major health concern and direct antiviral treatments are lacking. In this study, the authors employed an in silico approach to identify compounds inhibiting HEV replication. By using I-TASSER, they predicted the structure of the HEV methyltransferase protein and performed a virtual screen of compounds binding to the predicted binding site.

They further confirmed that the lead compound HPPA binds to the MTase and that it can inhibit HEV MTase activity in vitro and HEV replication in cellulo.

The manuscript is well written, the approach is appealing, and the findings have good potential. The identified compound HPPA has to be applied at a very high concentration to reduce HEV replication. While HPPA does not have very high chances to be developed into an anti-HEV therapy, this is an elegant proof-of-concept study which also has great potential to learn more about HEV biology. I have only some minor comments to further improve it:

For me it was not very clear how the authors predicted the HEV MTAse 3D structure. The way it is described in the material section, it appears that this had been done by “making use of the structural information of the resolved 3D structure” from the CHIKV MTase. And then they state that they used the CHIKV MTase because it is the “closest protein”. Is this based on the sequence homology or predicted structure? The number of conserved residues between the two enzymes as in Fig 2F appears small? How does it compare to the MTases of other tested viruses? What does not help is that I cannot find/download the supplementary material-did the authors forget to upload it?

How does the HEV MTAse 3D structure prediction by I-TASSER compares to using Alphafold?

For the cell-free assays the authors used a GT1-derived MTase but for in cellulo replication they used a GT3 strain. Please elaborate on this.

In Figure 9, panel A is confusing. The scale shows roughly a 30% decrease in HEV replication but in the legend and the result section it is stated a 3-log difference (3.2x10E6 copies to 4.3x 10E2.8 copies). A 30% inhibition is in better agreement with Figs 8 and 9C.

The drugs were almost applied immediately after transfection. With respect to the development of an anti-HEV regimen, it would be better to test the compound in a “curative” manner, meaning, applying it once replication has fully established e.g. earliest at 2 days post-transfection.

Some of the tables could be moved to supplementary material (e.g. Table 3 and 4)

Which anti-ORF2 antibody for both IF and WB was used? Give the appropriate information

Thorough editing is required. There a minor mistakes like double spaces (e.g line 68) or typos  (e.g. line 534 “Earlier, The identification” should be “Earlier, the identification”) throughout the manuscript

Generally, the language could be improved.

Reviewer 2 Report

Comments: MDPI_Viruses_1828368

Inhibition of Hepatitis E Virus Replication by novel inhibitor targeting Methyltransferase

As there are no approved drugs against HEV infection, patients are being treated with a broad spectrum of antiviral drugs, pegIFN2alpha and ribavirin (RBV) as monotherapy or combination therapy. In this study, the authors examined small molecule libraries and identified a lead drug candidate HPPA that may be of interest for antiviral activity against HEV. To identify potential inhibitors for HEV, the study screened 2,45,532 small molecule libraries using virtual in silico screening, modelling, and simulations against the refined model structure of HEV-MTase.  This study identified 10 hits or lead compounds that could be potential inhibitors of the HEV-MTase enzyme.  Subsequently, these 10 compounds were custom synthesized and screened for their inhibition of purified HEV-MTase activity.  Among these 10 compounds, HPPA was found to exhibit maximal inhibition of HEV-MTase activity and inhibition of viral replication in vitro without cytotoxic effects. Overall, the present study clearly indicates that HPPA could be a potential drug-like inhibitor against HEV virus.

Overall, the study is well designed and comprehensively presented, adequately referenced and concise in content. However, there are few minor corrections minor corrections in text, abbreviations and spacing throughout the manuscript that need to be addressed.

Suggestions

Line #20: “3, (4-hydroxyphenyl) propionic acid”, please abbreviate this here with HPPA as it has been introduced for the first time.

Lines #20, #27 and #72: The nomenclature “3, (4-hydroxyphenyl) propionic acid” used in abstract and keywords is not correct and please replace it with “3-(4-Hydroxyphenyl)propionic acid”. Also please correct wherever applicable.

Line 26: “PK/PD studies”, please elaborate PK/PD

Line #27: May be “Inhibition” may not a key word. suggest removing

Line #71-73: “Using above parameters, we report a lead molecule, 3, (4-hydroxyphenyl) propionic acid (HPPA) or Desaminotyrosine (DAT), which interacts and binds with MTase inhibiting HEV replication”. In this study, I didn’t see any data on about the Desaminotyrosine (DAT) molecule. If it is not relevant, please remove.

Line #149: “10mM Tris-Cl (pH 8.0), 100mM NaCl”, spacing is inconsistent. Some places it is written as “10 mM Tris-Cl” Please make necessary changes thought the manuscript.

Line #85: “(Supplementary data; Figure S1)” I didn’t see any supplementary data file supplemented with the manuscript.

Figure 1, Image not clear - please exchange with a sharper image

Please proofread and make necessary corrections wherever applicable
